# OpenReview forum: "3D-IntPhys: Towards More Generalized 3D-grounded Visual Intuitive Physics under Challenging Scenes"
_NeurIPS.cc/2023/Conference — NeurIPS 2023 poster_

### Official Review · Reviewer_SEVF · 2023-07-05

**Soundness:** 3 good
**Presentation:** 3 good
**Contribution:** 4 excellent
**Rating:** 7
**Confidence:** 4

**Summary:**

This paper proposed a combination of a NeRF-style perception module and a 3D point-based dynamics module to model challenging physical phenomena involving fluid, granular materials, and rigid objects. The perception module learns spatial-equivariant representations and transforms the environment into points. The dynamics module then exploits the compositional structure of the point systems with GNNs.

The proposed method significantly outperformed existing models that do not employ explicit 3D representations. The method also shows strong long-horizon prediction and generalization to complex scenarios.

**Strengths:**

The proposed method can model complex dynamics without strong supervision by GT 3D point trajectories, which are only available in simulated environments. It also generalizes well to unseen environments with extrapolated parameters due to its explicit point-based representation.

**Weaknesses:**

- This paper has omitted some details of the proposed method, including how the dataset is constructed, how each module is designed, how evaluation metrics are computed, and how particle velocities are computed. For contents that can be found in the supplementary material, a pointer in the main text would be better appreciated. For others, see Questions.
- In addition, since this paper claims that it can learn complex physical dynamics from purely visual supervision, it would be more interesting if there was a real-world demonstration of the proposed method.

**Questions:**

- The training of NeRF only promises the quality of object surfaces and does not impose any constraint on the interior of an object. As a result, the object occupancy field may only represent the surfaces of each object. I would assume that such surfaces are not good for modeling physical interactions, especially fluid-related ones. Why can your model reconstruct and predict the dynamics accurately?
- In Figure 2, does the Merge Loss refer to the spacing loss in Eq. 7?
- In section 4, how is the Structural Similarity Index Measure (SSIM) defined? What does it measure?
- The differences shown in Table 1 are impressive. But what are the units for MSE and SSIM?
- In Table 1, the GranularPush OoD results are significantly worse than the other cases. What are the reasons and implications for the difference?

**Limitations:**

The proposed method relies on point colors for segmentation, which implies that there must be a one-to-one mapping between color and instance, and the colors must be distinguishable. In addition, the fluid must be colored and non-transparent, which further limits the application of the proposed method.

---

> ### Author Rebuttal · Authors · 2023-08-10
>
>
>
> Thank a lot for identifying our work and providing valuable comments. Below, we try to address the raised questions:
>
>
>
> > The training of NeRF only promises the quality of object surfaces and does not impose any constraint on the interior of an object. As a result, the object occupancy field may only represent the surfaces of each object. I would assume that such surfaces are not good for modeling physical interactions, especially fluid-related ones. Why can your model reconstruct and predict the dynamics accurately?
>
>
> That is a good question. Actually, we are excited to find that, with the strong color prior imposed by conditional NeRF, the interior of fluids can be filled (refer to **additional pdf** in **General Responce** for more visualization results).
>
> The reason may be: if the NeRF model is only trained on one frame of pouring water, with high probability only the surface will have density. However, if we train it on bunch of videos of water pouring, the NeRF function f(color, position) will be strongly conditioned on color, which can predict the point as water if the point is colored blue from many views. We put more visualizations of the NeRF in the **additional pdf**.
>
>
> > In Figure 2, does the Merge Loss refer to the spacing loss in Eq. 7?
>
> Yes. We will rephrase it as spacing loss for better expression.
>
>
>
>
>
>
>
>
>
> > In section 4, how is the Structural Similarity Index Measure (SSIM) defined? What does it measure?
>
> The Structural Similarity Index Measure (SSIM) is a widely used image quality assessment metric. It is designed to quantify the similarity between two images, specifically evaluating the perceptual quality and structural information present in the images. SSIM was introduced as an improvement over traditional metrics like Mean Squared Error (MSE) or Peak Signal-to-Noise Ratio (PSNR), as it takes into account the human visual perception characteristics.
>
> The detailed definition can be found in [1].
>
>
> [1]: Zhou Wang and A. C. Bovik, "A universal image quality index," in IEEE Signal Processing Letters, vol. 9, no. 3, pp. 81-84, March 2002, doi: 10.1109/97.995823.
>
>
>
>
>
>
> > The differences shown in Table 1 are impressive. But what are the units for MSE and SSIM?
>
> The unit for MSE is pixel value squared, and the unit for SSIM is a fraction from 0 to 1.
>
>
>
> > In Table 1, the GranularPush OoD results are significantly worse than the other cases. What are the reasons and implications for the difference?
>
> First, Granular Push is a more challenging task because: (1) the diversity of fluids shapes is much larger than FluidPour and FluidCubeShake (at most time the fluids are restricted in the container), (2) the number of points is larger, making it easier to spread in the simulator.
>
> Among GranularPush, Granular OoD cases contain 12167 points, which is more than any other settings. The loss will be larger, but the results are still reasonable.

---

### Official Review · Reviewer_kxU1 · 2023-07-07

**Soundness:** 3 good
**Presentation:** 3 good
**Contribution:** 3 good
**Rating:** 5
**Confidence:** 3

**Summary:**

The paper proposes to learn intuitive physics from visual input by learning to infer the 3D scene structure and dynamics. The claim is that this explicit 3D reasoning generalizes better than existing implicit representations, and can be learned from videos without any explicit 3d supervision. To this end, the method uses PixelNeRF to learn the 3D representation of the scene and uses a graph-based neural network to predict the dynamical evolution of the scene. Evaluation is done in three simulated dataset and shows better performance compared to NeRF-dy and autoencoder models.

**Strengths:**

3D geometry and compositionality are key components for inferring physical dynamics, so the paper takes the right approach in enforcing 3D and compositionality in the model.

**Weaknesses:**

The paper does not show any results on real data. If the model is able to learn only from multiview videos the claims would have been stronger with real data.
Comparison with [16] is missing in the paper.


**Questions:**

How does the method perform on real-world videos?

**Limitations:**

The main limitations is it's unclear how the method will generalize to real-world scenarios.

---

> ### Author Rebuttal · Authors · 2023-08-10
>
> > The main limitations is it's unclear how the method will generalize to real-world scenarios.
>
> Thanks for your advice about real-world experiments. Our paper is among the first to embed explicit 3D information and strong inductive bias into 3D intuitive physics learning.
>
>
> Here we  want to emphasize that the work focuses more on learning complex visual dynamics from multi-view images, which itself is a challenging tasks, and there are only a few existing works. NeRF-dy is the closest to us, yet the model's generalization ability is limited. We have shown in the proposed work that we can significantly improve the generalization ability by operating with a hybrid of implicit and explicit in simulation datasets, as opposed to pure implicit, 3D representations.
>
> With stronger vision prior knowledge (object discovery, segmentation, e.g. SAM and SEER) in the future, we can try more exciting experiments in real world, making it possible to learn from bunch of videos around us.
>
>
>
> > If the model is able to learn only from multiview videos the claims would have been stronger with real data. Comparison with [16] is missing in the paper.
>
> Compared with Compnerf [16]: [16] only work on rigid objects and weakly-deformable ropes, most of the objects do not have topological changes, and they are all placed on the table. So  [16] uses object-centric dynamics model, which is not suitable for learning complex dynamics of fluids or granular materials. Our settings contain much more diverse 3D dynamics of challenging materials, which can not be solved using [16].

---

### Official Review · Reviewer_R6LT · 2023-07-08

**Soundness:** 3 good
**Presentation:** 3 good
**Contribution:** 2 fair
**Rating:** 6
**Confidence:** 2

**Summary:**

This paper proposes a method that learns to approximate physical interactions of primitive scenes from a set of videos through the use of of a NeRF and a dynamics prediction model. A static NeRF (based on PixelNeRF) is used to reconstruct a per time-step point cloud which is then used to supervise the dynamics prediction module via a Chamfer loss. A regularization is introduced in Eq. 7 of the paper to prevent the dynamic module to settle for a degenerate solution.

**Strengths:**

1) Qualitatively the proposed method outperforms prior art.

2) The paper is well written and motivated

**Weaknesses:**

1) No videos of interpolated or extrapolated dynamics are provided. MSE itself is not a good measure of the good a simulation is.



**Questions:**

How sensitive is the dynamics learning training to the parameter $\delta$ (line 153)? An ablation of a few values will aid in understanding the robustness of the system



**Limitations:**

Yes.

---

> ### Author Rebuttal · Authors · 2023-08-10
>
>
>
> Thank you for reviewing our work and providing valuable comments. Below, we try to address the questions you raised:
>
>
>
> > No videos of interpolated or extrapolated dynamics are provided. MSE itself is not a good measure of the good a simulation is.
>
>
>
> We kindly request the reviewers to watch the supplementary videos showcasing our results, which can be accessed through the anonymous link provided in Line:3 of supplement material of our submission. Your time and attention to these videos will provide valuable insights into the effectiveness and performance of our approach.
>
>
>
> > How sensitive is the dynamics learning training to the parameter $\delta$ (line 153)? An ablation of a few values will aid in understanding the robustness of the system
>
>
> That is a good question. In Line:58 of the supplementary material, we explain the choice of $\delta$: we select the threshold so that each point will have on average 20$\sim$30 neighbors. Through extensive experiments, we found that 20$\sim$30 neighbors is a sweet point: when $\delta$ is too small, powers from surrounding nodes are weakly propagated, when $\delta$ is too large, the interactions between points becomes redundant and is too slow to run.
>
> Following the advice from the reviewers, we do experiments on ablation on different $\delta$ value, and more results are put in the **additional pdf** of the **General Response** section.

---

> > ### Comment · Reviewer_R6LT · 2023-08-16
> > **Post-rebuttal update**
> >
> > I would like to thank the authors for the rebuttal. I apologize for missing the link to the videos in the supplementary and I thank the authors for pointing it out. The videos demonstrate that their method is able to generate simulations that are more physically plausible than prior-art. I also am satisfied with the ablation of $\delta$ and thank the authors for providing it. After looking at all the other reviewers and the rebuttal I have decided to raise my recommendation to a "Weak Accept"

---

### Official Review · Reviewer_493i · 2023-07-11

**Soundness:** 3 good
**Presentation:** 3 good
**Contribution:** 2 fair
**Rating:** 6
**Confidence:** 2

**Summary:**

Authors propose an approach to learn physics based model of real world objects from multi-view video. Authors argue that 3D representation of the scene is important to reason about the physics and use per-frame Nerf to extract the object point cloud from every frame of the multi-view video. They build a graph of the object points and use spatial message passing to learn the dynamics.
Authors use Nvidia FleX to generate simulated data for testing and compare their method with NeRF-dy (Li et al. CoRL'21).

**Strengths:**

+ The proposed method is simple and intuitive.
+ The idea of leveraging Nerf to build a 3D model of the world and using it to model physics is interesting.
+ The method compares favourably to baseline.

**Weaknesses:**

[Technical]
1. Sec 3: Proposed method consists of two steps, extracting the scene point cloud using Nerf [41] and computing dynamics using message passing [7,36,37].
Isn’t this similar to [35] where they predict 3D particle locations using NN and then use message passing + some other learning to predict the dynamics? If Nerf produces better points why can’t we replace the MLP based points from [35] with Nerf and use rest of the method? They also don’t make assumptions about physical properties of objects (also see pt. 2).
Is the key idea here that [35] requires GT 3D supervision whereas the proposed method does not? But doesn't having multi-view images with known cameras (as used in the method) sort of provide this supervision?

2. L154: what is captured in point attribute a_i^v? How is it obtained? Is it known beforehand? Adding dimensionality to variables would also help.
Unlike proposed method, prior work [35] does not require prior knowledge about the physical properties of the object or the scene. Isn’t the proposed method restrictive? Please correct me if I misunderstood this.

3. L141: How is object segmentation obtained? Is it provided as input? All inputs and outputs must be clearly explained.
A good idea would be to mention these in the intro to method section. Similarly all assumptions about the problem setting such as in L168-171 should also be clarified early on.

[Minor]
L106-109: Authors write they learn 3D-IntPhysics without any 3D supervision which is not entirely true. They use multi-view images with known camera for Nerf based representation which is also not entirely "not 3D".
Authors also say that prior work uses GT trajectories which are hard to obtain in real setting, but getting multi-view images with GT camera parameters is also not trivial.

**Questions:**

I'm not an expert in this area. I like the ideas presented here as they are simple and make intuitive sense. However I'm concerned about the novel contributions of the work. Please see pts. 1,2 above. I'd be happy to update my score after the rebuttal.

**Limitations:**

Authors briefly mention one of the limitations along with the conclusions but a more in-depth analysis would be appreciated. This could go in the supp. mat. and authors can at least leave a pointer in the main paper.

---

> ### Author Rebuttal · Authors · 2023-08-10
>
>
> Thank you for reviewing our work and providing valuable comments. Below, we try to address the questions you raised:
>
>
>
>
>
> > Proposed method consists of two steps, extracting the scene point cloud using Nerf [41] and computing dynamics using message passing [7,36,37]. Isn’t this similar to [35] where they predict 3D particle locations using NN and then use message passing + some other learning to predict the dynamics? If Nerf produces better points why can’t we replace the MLP based points from [35] with Nerf and use rest of the method? They also don’t make assumptions about physical properties of objects (also see pt. 2). Is the key idea here that [35] requires GT 3D supervision whereas the proposed method does not? But doesn't having multi-view images with known cameras (as used in the method) sort of provide this supervision?
>
> The settings of VPGL [35] differs a lot from our paper: (1) [35] requires GT 3D particle set as supervision to train the visual frontend, while our 3D-IntPhys does not need particle-level supervisions (2) In [35], the 3D representation is a particle set, which is an ordered list. As a result, it can only generate a fixed number of points. In contrast, our method produces dense representations that are not limited to ordered sets, making it more flexible and adaptable to systems of varying sizes. (3) [35] assumes that we have a learned dynamics model from the simulation as a dynamics prior, while we learn the dynamics model from the data.
>
> Simply replacing MLP in VGPL with NeRF will not work. First, the particle set  is  an ordered list, so that we can infer the physical properties using MLP directly. Also, [35] still needs to train a dynamics prior on dense 3D GT obtained from a simulator, which is not applicable in our settings, since we aim to directly learn from visual observations. Though VGPL do not make assumptions about the physical properties, it  makes more assumptions about the ordered particle sets and the dynamics prior to do inference about the physical properties in the scene.
>
>
>
>
> > L154: what is captured in point attribute a_i^v? How is it obtained? Is it known beforehand? Adding dimensionality to variables would also help. Unlike proposed method, prior work [35] does not require prior knowledge about the physical properties of the object or the scene. Isn’t the proposed method restrictive? Please correct me if I misunderstood this.
>
>
> The point/edge attributes $a^v$ and $a^e$ are one-hot features which are used to define the physical properties of points (e.g.  fluids are [0,0,1], rigid objects are [0, 1, 0], containers are [0, 0, 1]). These attributes are defined after the segmentation, which are widely used in point-based dynamics learners.
>
> As we mentioned in the last question, though VGPL do not make assumptions about the physical properties, it needs more assumptions about the 3D representation (ordered particle set) and the pre-trained dynamics prior on 3D GT point-clouds to do inference about the physical properties in the scene.
>
>
>
>
>
>
>
>
> > L141: How is object segmentation obtained? Is it provided as input? All inputs and outputs must be clearly explained. A good idea would be to mention these in the intro to method section. Similarly all assumptions about the problem setting such as in L168-171 should also be clarified early on.
>
>
> We thank the reviewers for pointing this out. The object segmentation in our paper is acquired using color informations, where the input is actually some prior information in forms of (color, object) pairs, e.g. (blue, water), (yellow, sand), (green, box). By leveraging this paired information, we effectively segment the objects from the scene.
>
>
> Yet, we want to emphasize that the work focuses more on learning complex visual dynamics from multi-view images, which itself is a challenging tasks, and there are only a few existing works. NeRF-dy is the closest to us, yet the model's generalization ability is limited. We have shown in the proposed work that we can significantly improve the generalization ability by operating with a hybrid of implicit and explicit, as opposed to pure implicit, 3D representations. Additionally, visual foundation models like SAM have demonstrated impressive performance in segmentation, achieving results nearly identical to our method, as discussed in our **general response**. Therefore, SAM could be considered an alternative to color-based segmentation.
>
>
>
>
> > [Minor] L106-109: Authors write they learn 3D-IntPhysics without any 3D supervision which is not entirely true. They use multi-view images with known camera for Nerf based representation which is also not entirely "not 3D". Authors also say that prior work uses GT trajectories which are hard to obtain in real setting, but getting multi-view images with GT camera parameters is also not trivial.
>
>
> In the table-top settings we considered in this paper, obtaining the camera parameters through calibration is not particularly hard to implement, and dataset of as much as eight cameras have be created [3].
>
> Yet, from 2D inputs combined with camera parameters to 3D representations is still a 2D-to-3D lift [1]. Also, some work has demonstrates the potential to release the camera poses from learning NeRF   [2].
>
>
>
> [1] Visual Reinforcement Learning with Self-Supervised 3D Representations.
>
> [2] MonoNeRF: Learning Generalizable NeRFs from Monocular Videos without Camera Poses.
>
> [3] DexYCB:: A Benchmark for Capturing Hand Grasping of Objects
>
>
>
>
>
> > Authors briefly mention one of the limitations along with the conclusions but a more in-depth analysis would be appreciated. This could go in the supp. mat. and authors can at least leave a pointer in the main paper.
>
>
> Thanks for your advice, we will include a more detailed discussion about the limitation of our work and add a pointer in the main paper.

---

> > ### Comment · Reviewer_493i · 2023-08-15
> > **Post rebuttal update**
> >
> > Thanks authors for the rebuttal. It was very helpful. I've updated my score to 'weak accept'. In case of acceptance I urge the authors to improve the clarity a bit, eg. before the rebuttal I couldn't figure out that object properties are encoded as one-hot vectors in a^v. Similarly segmentation was also a bit unclear. Distinguishing this work from prior work would also be helpful for a wider audience.

---

> > > ### Author Response · Authors · 2023-08-15
> > > **Thanks for your valuable feedbacks**
> > >
> > > Thank again you for the valuable feedback. We will add the related clarifications and comparison with existing works to our revision, please let us know if you have further suggestions!

---

### Official Review · Reviewer_cs73 · 2023-07-23

**Soundness:** 3 good
**Presentation:** 3 good
**Contribution:** 3 good
**Rating:** 6
**Confidence:** 2

**Summary:**

The paper introduces a 3D point-based dynamic model for visual intuitive physics reasoning. Specifically, The model adopts a conditional NeRF as the perception module to extract 3D point clouds from multi-view images. It then employs a graph neural network that operates on the extracted points for forward dynamic predictions.

The proposed approach achieves state-of-the-art performance across three simulation tasks (FluidPour, FluidCubeShake, GranularPush) i, with evident improvement in both visual reconstruction quality and dynamic prediction accuracy.

**Strengths:**

The paper improves upon prior work in the following aspects:
1. The explicit 3D point-based representation gives a good inductive bias for reasoning about local states.
2. The use of conditional NeRF enables better generalizability to out-of-distribution scenarios compared to the prior art.
3. The proposed method predicts correct particle arrangement, achieving state-of-the-art results on the benchmark.

Compared to the image-based encoding that suffers from geometry inconsistency, the proposed 2D-to-3D lifting extracts and operates directly on explicit 3D representation, circumventing the inconsistency problem. The direct reasoning on consistent and local geometric structures enables the method to achieve better accuracy on dynamic predictions.

Overall, the paper is well-written, the idea is sound, and the results back up the design choices.

**Weaknesses:**

While the work has several merits, it also has some limitations/issues:
- Reliance on instance-level masks. While SAM can provide instance-level masks (as in Supp. 148-152), how the mask quality can impact the performance
- The acquisition of point attributes a^v and relation a^e is not described. If these features have to be defined manually, then it should also be listed as one of the limitations.
- An additional stage of training for the conditional NeRF is required. Can the proposed approach work on point clouds extracted from Structure from Motions? If a^v and a^e are not associated with perceptual features, then the PixelNeRF can be replaced with something simpler.

**Questions:**

Most of the questions I have are listed in the Weakness section, and I am willing to adjust my scores accordingly based the authors' responses and other reviewers' comments.

Some additional minor comments:
- Figure 6.: could be improved with x- and y-labels. I assume x is the horizon, and y is the chamfer loss.

Post-rebuttal update: The rebuttal answered most of my questions and concerns, and also clarified the issues raised by other reviewers. So I stand by my original assessment -- weak accept --- for this submission. Overall the idea and execution are very neat, and I would love to see more follow up work on this one.

**Limitations:**

The paper includes potential limitations and societal impact in the conclusions and the supplementary.

---

> ### Author Rebuttal · Authors · 2023-08-10
>
>
> Thanks for the careful review and valuable comments. Here we will address the questions raised up by the reviewer:
>
>
>
> > Reliance on instance-level masks. While SAM can provide instance-level masks (as in Supp. 148-152), how the mask quality can impact the performance
>
>
>
> The instance-level masks in the paper are acquired solely from color information, which are not acquired from ground-truth masks. In the **additional pdf file** in General Response, we demonstrate that the masks we use still have unmarginal difference from ground-truth masks, possibly due to points with ambiguous colors. However, despite this challenge, our pipeline can still effectively learn reasonable dynamics using the same input. Meanwhile, to make it a fair comparison, we also provide the color segmentation as input in baseline methods like [34].
>
> Further we did experiments on mask quality to show that our method is robust to the segmentation quality: we perturb the segmentation by 1% and 5% on the Env-1 settings of FluidPour:
>
> |    | Loss@T= 5 |Loss@T=20 | Loss@T=35|
> | ---- | ---- | ----- | ----- |
> | Perturb 0%   |  0.020  | 0.030   | 0.020   |
> | Perturb 1% | 0.020   | 0.040   |0.030   |
> | Perturb 5% | 0.020   | 0.045   |0.028   |
>
>
>
>
>
> As for SAM, we compare the SAM segmentation as a strong visual prior on FluidPour and find that it can effectively segment the water out, which means that we can use  SAM on our scenes as an alternative. We sample 10 frames and 3 views from each settings and compute the IoU (%) of SAM and our color-based segmentation of water:
>
> |    | Env-1|Env-2|Env-3|Env-4|Env-5|
> | ---- | ---- | ----- | ----- | ----- | ----- |
> | View-1   |  100  | 100   |100   |100   | 98 |
> | View-3 | 99   | 99   |98   | 99   | 96 |
> | View-5 | 100   |100   |100   |100   | 98 |
>
>
>
> > The acquisition of point attributes a^v and relation a^e is not described. If these features have to be defined manually, then it should also be listed as one of the limitations.
>
>
>
> The point attributes $a^v$ and $a^e$ are one-hot features which are used to define the physical properties of points (e.g. fluid or rigid). These attributes are defined after the segmentation, which are widely used in point-based dynamics learners. While  [35] can learn physical properties automatically, it needs a trained dynamics prior on GT 3D particle sets and have stronger assumption about the 3D representation as ordered particle sets.
>
> Yet, we want to emphasize that the work focuses more on learning complex visual dynamics from images, as opposed to solving object discovery and feature discovery in general.  Learning fluids dynamics from videos is a challenging task, and there are only a few existing works. NeRF-dy is the closest to us, yet the model's generalization ability is limited. We have shown in the proposed work that we can significantly improve the generalization ability by operating with a hybrid of implicit and explicit, as opposed to pure implicit, 3D representations. 	The future investigation may include using LLM to provide dynamics prior knowledge giving the physical properties of the objects (e.g. fluids or rigid body).
>
>
>
>
>
> > An additional stage of training for the conditional NeRF is required. Can the proposed approach work on point clouds extracted from Structure from Motions? If a^v and a^e are not associated with perceptual features, then the PixelNeRF can be replaced with something simpler.
>
>
> We need an additional stage of training conditional NeRF, which is actually effective: we only trained on a small subset of videos ($\sim$  50), and the conditional NeRF can generalize to the rest of the scenes ($\sim$ 450) in the dataset, which shows that conditional NeRF is effective and can generalize well to extrapolated settings.
>
> On the other hand, Structure from Motion (SfM) primarily produces the surface representation of point clouds using multi-view inputs, which may not be ideal for learning fluid dynamics. Conditional Neural Radiance Fields (NeRF) are trained on a bunch of videos including thousands of different frames, enabling them to learn valuable biases and effectively reconstruct the inner points of fluids.
>
> Furthermore, it is important to note that SfM offers a sparse representation of the scene, whereas NeRF provides dense reconstruction. This characteristic makes NeRF more versatile, allowing it to handle sparse sampling with greater flexibility.
>
>
>
> > Figure 6.: could be improved with x- and y-labels. I assume x is the horizon, and y is the chamfer loss.
>
> Yes, x is horizon and y is the chamfer loss of predicted states. We will add the detailed x, y labels for better understanding.

---

> > ### Comment · Reviewer_cs73 · 2023-08-13
> > **Thanks for the comprehensive response**
> >
> > I have read the authors' responses and the other reviewers' comments. My concerns are addressed adequately, and I am inclined to stand by my original recommendation, i.e., Weak Accept. But for now, I'd like to see how other reviewers think and keep track of the follow-up discussions.

---

### Author Rebuttal · Authors · 2023-08-10

# General Response

We thank the reviewers for their valuable comments on our paper. We are excited to see that reviewers (Reviewer cs73, 493i, R6LT, kxU1, SEVF) identified the novelty of our technical contribution, acknowledged the better performance of our methods over baselines (Reviewer cs73, 493i, R6LT, kxU), and found the paper well-written and motivated (Reviewer cs73, R6LT).



Here we try to address some commonly asked questions:



> Comparison with existing SOTA of 3D intuitive physics [34], [35] and [16] (Reviewer 493i)

- Compared with nerf-dy [34]: our method use explicit 3D representation instead of implicit representation, which we show in our paper that our method can generalize better than [34].
---
- Compared with VPGL [35]: (1) [35] requires GT 3D particle set as supervision to train the visual frontend, while our 3D-IntPhys does not need particle-level supervision (2) In [35], the 3D representation is a particle set, which is an ordered list. As a result, it can only generate a fixed number of points. In contrast, our method produces dense representations that are not limited to ordered sets, making it more flexible and adaptable to systems of varying sizes. (3) [35] assumes that we have a learned dynamics model from the simulation as a dynamics prior, while we learn the dynamics model from the data.
---

- Compared with Comp_nerfdyn [16]: [16] only works on rigid objects and a rope that exhibits only slight deformation, most of the objects do not have topological changes, and they are all constrained to move in a 2D plane. So while the object-centric dynamics model used in [16] can solve the tasks in their paper, the object-centric representation is not suitable for learning the complex dynamics of fluids or granular materials, as in our paper. Our settings contain much more diverse 3D dynamics of challenging materials, which can not be solved using [16].



>  How to get the point attributes $a^v, a^e$ (Reviewer cs73, 493i)

The point attributes $a^v$ and $a^e$ are one-hot features which are used to define the physical properties of points (e.g. fluid or rigid). These attributes are defined after the segmentation, which are widely used in point-based dynamics learners.



> About color segmentation. (Reviewer R6LT, 493i)


We want to emphasize that the work focuses more on learning complex visual dynamics from images, as opposed to solving object discovery and segmentation in general. Learning fluid dynamics from videos is a challenging task, and there are only a few existing works. NeRF-dy is the closest to us, yet the model's generalization ability is limited. We have shown in the proposed work that we can significantly improve the generalization ability by operating with a hybrid of implicit and explicit, as opposed to pure implicit, 3D representations. We agree object segmentation is a critical visual understanding problem, and solving it is an important next step to getting a more general visual dynamics learning framework.

Also, we noticed that stronger visual foundation models like SAM and SEER are proposed as excellent segmentation tools. We further do experiments to show that it is possible to use SAM as an alternative, specifically, we compare  SAM on FluidPour environments and find that it can effectively segment the water out. We sample 10 frames and 3 views from each setting and compute the IoU (%) of SAM and our color-based segmentation of water:

|    | Env-1|Env-2|Env-3|Env-4|Env-5|
| ---- | ---- | ----- | ----- | ----- | ----- |
| View-1   |  100  | 100   |100   |100   | 98 |
| View-3 | 99   | 99   |98   | 99   | 96 |
| View-5 | 100   |100   |100   |100   | 98 |



> About the video visualization (Reviewer R6LT)

We kindly refer the reviewer to the video submitted as a link in the supplementary material (Line 3). The video was uploaded anonymously before the submission deadline.

---

### Decision · Program_Chairs · 2023-09-21

**Decision:**

Accept (poster)

**Comment:**

This paper appears was unanimous accept by the reviewers. They all appreciated the novelty and improvement over baselines, found the paper well-written and motivated, and felt the paper is a good contribution to the field.  The author rebuttal was reviewed and commented on by all with healthy discussion between the authors and the reviewers and the AC.

Congratulations!

In revising your paper for the camera-ready, please include and comments and clarifications that were made or proposed during the rebuttal and discussion phase.